# Dispersion of LPG from spherical storage tanks: Power-law scaling and comparative analysis of LFL vs. 50% LFL

Keyvan Sarebanzadeh[1], Mahboubeh Es'haghi 📵[1,2]*

1 Department of Occupational Health Engineering and Safety at Work, Faculty of Public Health, Kerman University of Medical Sciences, Kerman, Iran, 2 Student Research Committee, Kerman University of Medical Sciences, Kerman, Iran

* m.eshaghi@kmu.ac.ir, esaghij@gmail.com

## Abstract

Liquefied Petroleum Gas (LPG) is typically stored in pressurized spherical tanks, where accidental leaks can create dense, flammable vapor clouds. This study used PHAST to simulate LPG dispersion from a refinery-scale spherical tank, considering various leak diameters (5–805 mm), leak locations, seasonal meteorological conditions, and three propane–butane mixtures (15/85, 30/70, and 50/50 by volume). Dispersion distances were evaluated at both the Lower Flammability Limit (LFL) and 50% LFL thresholds. The findings indicate that leak diameter is the primary factor influencing dispersion extent, showing strong correlations for both LFL ($\rho = 0.89$) and 50% LFL ($\rho = 0.91$). Predicted dispersion distances downwind ranged from approximately 20–60 m for small leaks to around 400–800 m for larger releases, depending on the concentration threshold and release conditions. Distances at the 50% LFL were consistently greater than those at the LFL. Power-law regression revealed nearly linear scaling between dispersion distance and leak diameter ($b = 0.94$ for LFL and $b = 0.96$ for 50% LFL), explaining over 80% of the observed variance. Butane-rich mixtures resulted in longer dispersion distances at the LFL, while compositional effects were not significant at the 50% LFL. Meteorological and temporal factors had limited influence under typical site conditions. Overall, the results emphasize comparative scaling behavior rather than pointwise concentration prediction and demonstrate deviations from ideal D² scaling due to turbulence, buoyancy, and atmospheric entrainment. Using both LFL and 50% LFL thresholds provides a conservative and practically relevant basis for hazard zoning, quantitative risk assessment, and emergency planning at LPG storage facilities.

## Introduction

Liquefied Petroleum Gas (LPG), primarily composed of propane ($C_3H_8$) and butane ($C_4H_{10}$), is commonly stored in pressurized spherical tanks due to its high energy

provided the original author and source are credited.

**Data availability statement:** all data are mentioned in the manuscript.

**Funding:** The author(s) received no specific funding for this work.

**Competing interests:** No authors have competing interests.

density, compact footprint, and favorable handling characteristics. In the event of an accidental release, LPG can quickly vaporize and form dense, ground-hugging clouds that may travel significant distances before ignition, posing serious risks to personnel, equipment, and surrounding communities [1–3]. Accurately determining the hazardous extent of these vapor clouds is essential for land-use planning, facility siting, emergency preparedness, and quantitative risk assessment in refinery environments.

In consequence modeling, the Lower Flammability Limit (LFL) is traditionally used as the primary criterion to delineate flammable vapor cloud boundaries. However, several regulatory and industrial guidelines, including those issued by the Center for Chemical Process Safety (CCPS), the National Fire Protection Association (NFPA), and the American Petroleum Institute (API), recommend the use of sub-LFL criteria such as 50% LFL to account for turbulence-induced concentration fluctuations, spatial variability, and uncertainties inherent in dispersion modeling [4–8]. These sub-LFL thresholds are intended to provide more conservative safety margins, particularly under conditions where localized concentrations may transiently exceed the ignition limit even when average values remain below the LFL.

Despite the widespread application of dispersion models such as SLAB, DEGADIS, ALOHA, and PHAST in industrial risk assessments [4,5,9–13], there have been few studies focusing on LPG releases from pressurized spherical storage tanks operating under refinery conditions. Additionally, there is a lack of systematic comparisons between LFL and 50% LFL dispersion distances in the existing literature, even though the selection of the threshold directly influences hazard zoning and emergency planning decisions. Existing studies typically provide deterministic dispersion distances for specific scenarios and rarely use statistical methods to assess parameter influence, uncertainty, or scaling behavior across a wide range of operational conditions.

To address these gaps, the present study conducts a comprehensive dispersion analysis of LPG releases from a refinery-scale spherical storage tank using PHAST. Various leak diameters, LPG compositions, and typical meteorological conditions are considered. Dispersion distances are assessed separately at both the LFL and 50% LFL thresholds. Statistical analyses, including correlation analysis, hypothesis testing, and power-law regression, are utilized to quantify the influence of key parameters and establish practical scaling relationships relevant to engineering practices.

This study contributes to the field of LPG dispersion analysis in several ways. Firstly, it systematically compares dispersion behavior at both the LFL and 50% LFL thresholds for pressurized spherical LPG storage tanks under refinery operating conditions. Secondly, it develops statistically supported power-law correlations to quantify the scaling relationship between leak diameter and dispersion distance. Thirdly, the influence of propane–butane composition on dispersion extent is evaluated at different flammability thresholds. Additionally, it derives practical empirical relationships to support refinery-scale hazard zoning and quantitative risk assessment. Finally, the study clarifies the conservative implications of adopting sub-LFL criteria, such as 50% LFL, in industrial safety and land-use planning applications.

The remainder of this paper is organized as follows. The literature review section summarizes previous studies on dense-gas dispersion modeling, flammability thresholds, and LPG release studies. The material and methods section describes the modeling framework, scenario definition, assumptions, and statistical methodology. The results section presents the dispersion outcomes and regression analyses. The discussion section interprets the findings in relation to previous studies, practical implications, and limitations. Finally, the conclusion section summarizes the main findings and outlines directions for future research.

## Literature review

Extensive research has been conducted on dense-gas dispersion through large-scale experimental programs, most notably the Burro, Coyote, and Maplin Sands field trials. These experiments have established the fundamental influence of source strength, atmospheric stability, and buoyancy deficit on the evolution of heavy vapor clouds, forming the empirical basis for many dispersion models used in industrial safety analysis [14–17].

Based on experimental datasets, integral and semi-empirical dispersion models such as SLAB, DEGADIS, HEGADAS, and related formulations have been widely adopted in regulatory and industrial applications due to their computational efficiency and ease of implementation. However, Model Evaluation Protocol (MEP) studies have shown that such models often under-predict near-field concentrations while over-predicting far-field dispersion, especially under stable atmospheric conditions [18–21].

PHAST, which incorporates the Unified Dispersion Model (UDM), is one of the most widely used industrial tools for consequence analysis. The UDM integrates jet, dense-gas, and passive dispersion regimes within a single framework and has been validated against several large-scale experimental datasets. However, comparisons with Computational Fluid Dynamics (CFD) models indicate that PHAST may underestimate peak concentrations in the immediate vicinity of the release due to simplified turbulence representation, particularly for high-momentum or large-diameter releases [4,10,22].

CFD-based models, including FLACS, ANSYS Fluent, and OpenFOAM, provide higher spatial and temporal resolution and are increasingly being used to investigate dense-gas dispersion in complex scenarios. Several CFD studies have indicated that theoretical scaling laws, such as the $D^2$ relationship from orifice flow theory, often overestimate dispersion distances for large releases. Instead, a consistent observation of sub-linear scaling behavior suggests the mitigating effects of turbulence, atmospheric entrainment, and buoyancy-driven dilution [10,22,23].

Regarding safety thresholds, most academic dispersion studies typically define hazardous cloud boundaries exclusively based on the LFL. However, guidance from industrial organizations such as API, NFPA, and CCPS is increasingly suggesting the use of sub-LFL criteria, such as 50% LFL, as more conservative design thresholds. Despite this recommendation, there is a lack of systematic academic comparisons between LFL and 50% LFL dispersion distances [7,8]. Existing studies have primarily focused on pipelines, transportation vessels, or urban releases, with limited attention given to pressurized spherical LPG storage tanks operating under refinery conditions [24–26].

The influence of LPG composition has been addressed in a few studies. Available evidence suggests that butane-rich mixtures tend to produce more persistent, ground-hugging vapor clouds, while propane-rich mixtures dilute more rapidly due to higher volatility. However, there is a lack of quantitative assessments comparing compositional effects at different flammability thresholds, particularly at both LFL and 50% LFL levels, in the literature. Additionally, while statistical techniques such as correlation analysis, hypothesis testing, and regression-based scaling have been recommended to improve dispersion assessment rigor, they are rarely used in LPG dispersion studies. Most published works report deterministic dispersion distances without systematically evaluating parameter influence, uncertainty, or scaling behavior.

In summary, the existing literature highlights several key gaps: (1) limited research on refinery-scale spherical LPG storage tanks; (2) a lack of systematic comparison between LFL and 50% LFL thresholds; (3) insufficient statistical evaluation of the influence of leak diameter, composition, and operating conditions; and (4) a scarcity of empirical scaling correlations

applicable to refinery-specific scenarios. These gaps motivate the integrated modeling and statistical framework adopted in the present study. Table 1 summarizes the most relevant prior studies, highlighting their modeling approaches, safety thresholds, key findings, and limitations.

## Materials and methods

This study evaluates the dispersion behavior of LPG released from a pressurized spherical storage tank located at an industrial refinery in Iran. The tank has a nominal capacity of 20,000 bbl (3180 m³) and normally operates at 40 °C and 8.2 barg, with a design pressure of 18.8 barg. LPG is stored under subcooled conditions; therefore, rapid flashing occurs immediately upon depressurization. Although PHAST performs VLE calculations internally, the expected flash fraction at operating conditions was independently verified using REFPROP correlations to confirm that the stored mixtures remain fully subcooled and generate a high vapor fraction upon release. This verification supports the applicability of dense-gas dispersion modeling as implemented in PHAST's UDM. Fig 1 illustrates the schematic configuration of the spherical tank and leak locations.

The principal design and operating characteristics of the storage tank are summarized in Table 2. The LPG stored in the tank consists of propane and butane blended in different proportions depending on seasonal supply conditions. Three representative compositions, 15/85, 30/70, and 50/50 propane–butane by volume, were selected based on regional specifications and standard recommendations for consequence analysis.

Five circular leak diameters (5, 25, 50, 100, and 805 mm) were analyzed. Leak diameters of 5, 25, 50, and 100 mm represent typical refinery failure mechanisms, including corrosion pinholes, small-bore piping failures, gasket or flange degradation, and partial nozzle damage, as reported in CCPS and API failure-frequency databases and related studies [39–42]. The 805-mm diameter corresponds to the internal diameter of the tank's main outlet nozzle, as confirmed by mechanical design documentation. Although a full-bore rupture is a low-probability event, it is routinely included in quantitative risk assessments as a Maximum Credible Event (MCE) to define upper-bound hazard distances for emergency planning and land-use zoning. Both bottom and top leak locations were considered. Bottom leaks represent physically plausible failures of the outlet line, while top leaks were included as complementary sensitivity cases to isolate the influence of release elevation rather than as dominant failure scenarios.

Meteorological conditions were determined from long-term site-specific records; including air temperature, ground surface temperature, wind speed, relative humidity, and atmospheric stability class for both daytime and nighttime. Two seasonal periods (spring–summer and autumn–winter) commonly used in refinery consequence analysis were taken into account. Neutral (D) and stable (E/F) stability classes were selected; with stable nighttime conditions and low wind speeds specifically included due to their tendency to result in the longest dispersion distances for dense-gas releases. Ground surface temperature data was not directly available and was therefore estimated using a standard correlation commonly used in consequence modeling in Equation 1 [43]. The meteorological conditions used in the simulations are summarized in Table 3.

$$T_{air} = 14.6 + 0.44 \times SRT \tag{1}$$

Dispersion distances were quantified using two safety-relevant concentration thresholds: the LFL and 50% LFL. The LFL represents the minimum vapor concentration capable of sustaining ignition and is a fundamental parameter in consequence analysis, typically determined using the Le Chatelier mixing rule [44,45]. The 50% LFL threshold was included as a conservative criterion recommended in several safety guidelines to account for concentration fluctuations, turbulence, and spatial variability that may locally exceed the ignition limit even when average concentrations remain below the LFL. In this study, LFL and 50% LFL were treated as independent analytical endpoints rather than alternative predictive criteria, and separate datasets were generated for each threshold.

**Table 1. Summary of key studies on LPG dispersion.**

| Author(s), Year | Modeling Tool | Variables Studied | Threshold (s) Used | Key Findings | Limitations |
|---|---|---|---|---|---|
| Luketa-Hanlin, Koopman, and Ermak (2007) [27] | CFD | LNG, turbulence model, atmospheric conditions, obstacles | LFL | Discussed best practices for CFD modeling of LNG dispersion and validation against Burro and Falcon tests | Supports comparison between integral models and CFD approaches |
| Pontiggia et al. (2009) [28] | CFD (FLACS) | Leak diameter, wind, urban geometry | LFL | Turbulence reduces cloud growth compared to D² scaling | Focused only on LFL; no statistical validation |
| Cormier et al. (2009) [29] | CFD + medium-scale experiments (BFTF) | Pool size, evaporation rate, wind speed, turbulence intensity | LFL | provides conservative exclusion zones; source term dominates at low wind | Direct justification for using LFL |
| Pontiggia et al. (2011) [22] | CFD | LPG release, Leak diameter, release rate, turbulence, entrainment | LFL | Showed sub-linear scaling due to turbulence and entrainment | Computationally intensive |
| Sun et al. (2013) [30] | CFD validated with Burro tests | LNG, Leak location, terrain, wind speed, obstacles | LFL | CFD predictions showed improved accuracy over DEGADIS, with <20% error | Confirms limitations of integral models and need for empirical scaling |
| Li et al. (2017) [31] | CFD + engineering model | Methane (confined space)Leak size, time evolution, flammable region geometry | LFL | Developed regression-based engineering correlations | Supports statistical scaling methodology |
| Luo et al. (2018) [32] | Multiphase CFD | LNG, Phase change, pool spreading, impoundment geometry | LFL/ half-LFL | Multiphase CFD improved prediction of pool formation and vapor dispersion | Reinforces importance of validated dispersion modeling |
| Giannissi and Venetsanos (2019) [33] | CFD | Cryogenic $H_2$ and LNG, Release momentum, buoyancy, humidity, Froude number | LFL | Showed sub-linear scaling of dispersion distances and strong influence of momentum vs buoyancy | Supports deviation from ideal D² scaling law |
| NFPA (2019) [8] | Guidelines | Fire safety standards | LFL + 50% LFL | Endorses conservative thresholds | No validation with LPG dispersion |
| Àgueda et al. (2020) [34] | Simulation/ integral model (FLACS etc) | LNG & propane pool dispersion, ambient conditions (wind, temperature, humidity) | Both 100% LFL and 50% LFL (via DSF50) | Provides boxplots and comparisons of distances to both 100% (LFL) and ~50% LFL; shows that empirical downwind lengths for 50% LFL are consistently lower than LFL | Mostly pool/ LNG release cases; not LPG mixture or spherical tank; limited geometries |
| Lyu et al. (2023) [35] | CFD (urban accident case) | Droplet effects, terrain, large obstacles, urban geometry | LFL | Droplets enlarge LFL-affected area by ~1.9–2.1×; obstacles and terrain strongly alter dispersion patterns | Only LFL considered; case-specific (tank truck accident); no 50% LFL; focused on instantaneous release |
| Le et al. (2023) [5] | CFD | Leak diameter, wind speed (50/50 mix) | LFL | Leak diameter & wind strongly influence dispersion | Single composition; no sub-LFL consideration |
| Xiao et al. (2023) [36] | Lab experiments + numerical | Offshore transient releases | LFL | Good agreement between experiment & model | No statistical tests applied |
| Lozano et al. (2024) [10] | CFD (FLACS-CFD) | Leak properties, wind, road environment | LFL | Quantified gas cloud extension; road obstacles affect dispersion | No 50% LFL; tanker scenario |
| Duong and Kang (2025) [37] | CFD/ numerical | Leak diameter, mixture composition, pipeline leaks | LFL | Butane-rich mixtures produce wider dispersion distances | Pipeline scenario; no 50% LFL |
| Yuan et al. (2025) [38] | CFD/ numerical | Ventilation, wind speed, indoor geometry | LFL | Ventilation reduces cloud volume; logarithmic relation with wind speed | Indoor-focused; not spherical storage |
| Present study | PHAST + statistics | Leak diameter, mixture composition, atmospheric stability | LFL + 50% LFL | Shows near-linear scaling and statistical tests | Extends prior work with dual thresholds + statistical validation |

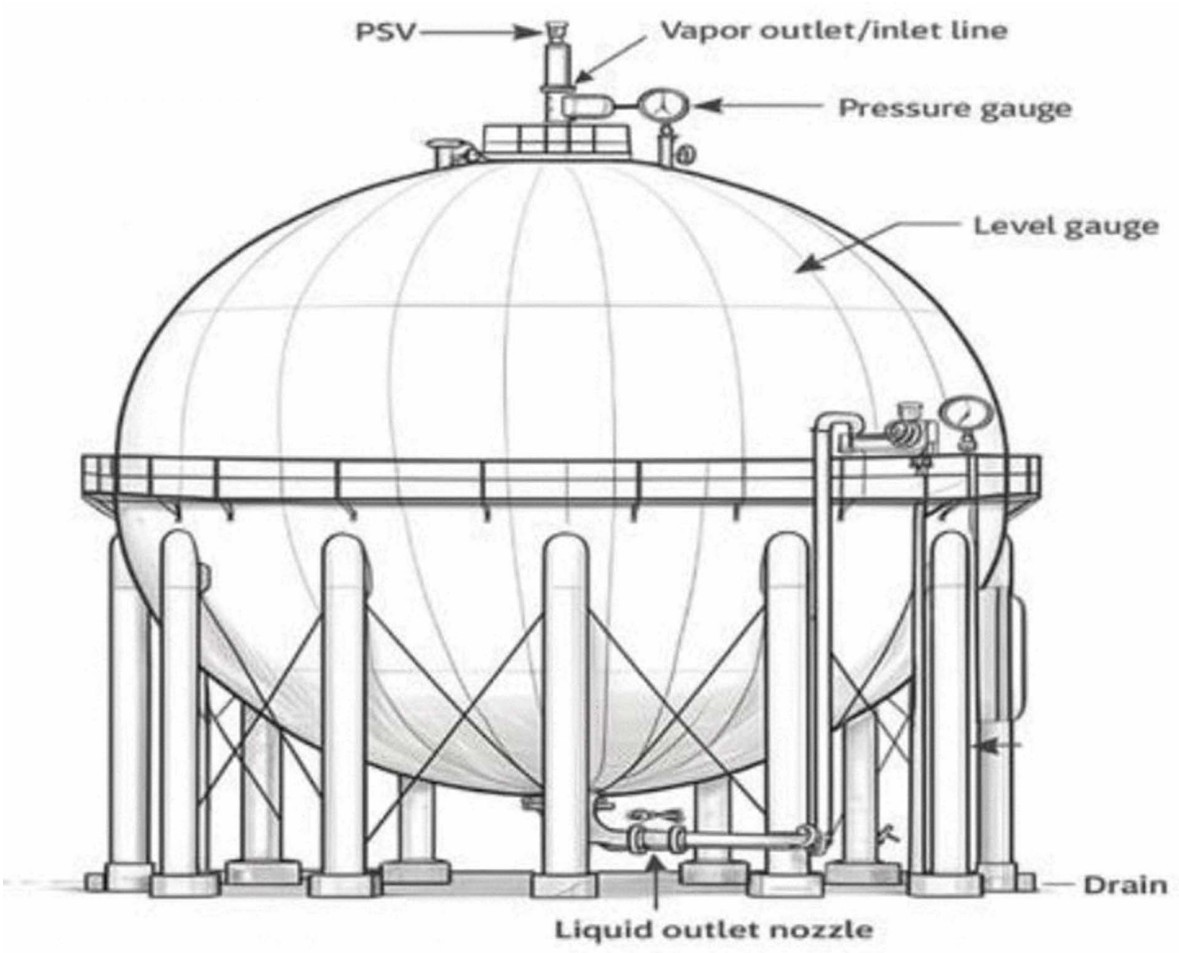

**Fig 1. Schematic of spherical LPG tank and leak points.**

**Table 2. Specifications of the LPG spherical tank.**

| Material | Shape | Diameter (m) | Pressure (barg) | | Temperature (°C) | | Capacity | |
|---|---|---|---|---|---|---|---|---|
| | | | Operational | Design | Operational | Design | m³ | bbl |
| LPG | Spherical | 19.4 | 8.2 | 18.8 | 40 | 85 | 3180 | 20000 |

**Table 3. Meteorological conditions used for dispersion simulations.**

| Period | Time | Avg. Relative Humidity (%) | Avg. Wind Speed (m/s) | Avg. Ground Temperature (°C) | Avg. Air Temperature (°C) | Stability Class |
|---|---|---|---|---|---|---|
| First half of the year | Day | 58/1 | 5 | 45 | 34/4 | D |
| | Night | 58/9 | 2/8 | 36/5 | 30/7 | F |
| Second half of the year | Day | 56/3 | 4/2 | 23/9 | 25/1 | E |
| | Night | 71/2 | 2/8 | 12/2 | 20 | F |

Fig 2 presents a schematic three-dimensional illustration of a spherical LPG storage tank experiencing a bottom leak with a nozzle diameter of 50 mm. The released vapor cloud disperses in the wind direction (+X axis). Two conceptual concentration envelopes are shown: the LFL boundary (blue wireframe) and the broader 50% LFL boundary (green wireframe). These geometries are purely illustrative and are not derived from PHAST output. They are included solely to visualize the relative spatial extent of the two flammability thresholds.

After defining leak scenarios and concentration thresholds, raw dispersion outputs were exported from PHAST and preprocessed before statistical analysis. The dataset included a fully crossed combination of leak diameter (5 levels), LPG composition (3 levels), leak location (top and bottom), and meteorological condition (4 cases). A total of 240 deterministic dispersion simulations were conducted, with 120 cases evaluated at the LFL and 120 cases at the 50% LFL threshold. Each simulation represented a unique combination of parameters, with no repeated stochastic runs for identical scenarios. Therefore, all observations were treated as independent data points.

The relationship between leak diameter and dispersion distance (Eq. 2) was analyzed using a power-law formulation commonly applied in gas dispersion and release scaling studies [10,22]:

$$\ln(L) = \ln(a) + b{\cdot}\ln(D) \tag{2}$$

Where L (m) is the dispersion distance, D (mm) is the leak diameter, 'a' is a scaling coefficient, and 'b' is the scaling exponent describing the sensitivity of dispersion distance to changes in leak diameter. Ordinary Least Squares (OLS) regression was applied to the log-transformed data to estimate model parameters. Ninety-five percent confidence intervals (CI) and prediction intervals (PI) were calculated to quantify parameter uncertainty and dispersion variability.

Normality of dispersion distance data was assessed using the Shapiro–Wilk test [46], which indicated significant deviations from normality for both LFL and 50% LFL datasets. Therefore, non-parametric statistical methods were used for hypothesis testing. Spearman's rank correlation coefficient was used to evaluate monotonic relationships between leak diameter, LPG composition, and dispersion distance [47]. Mann–Whitney U tests were used for categorical comparisons,

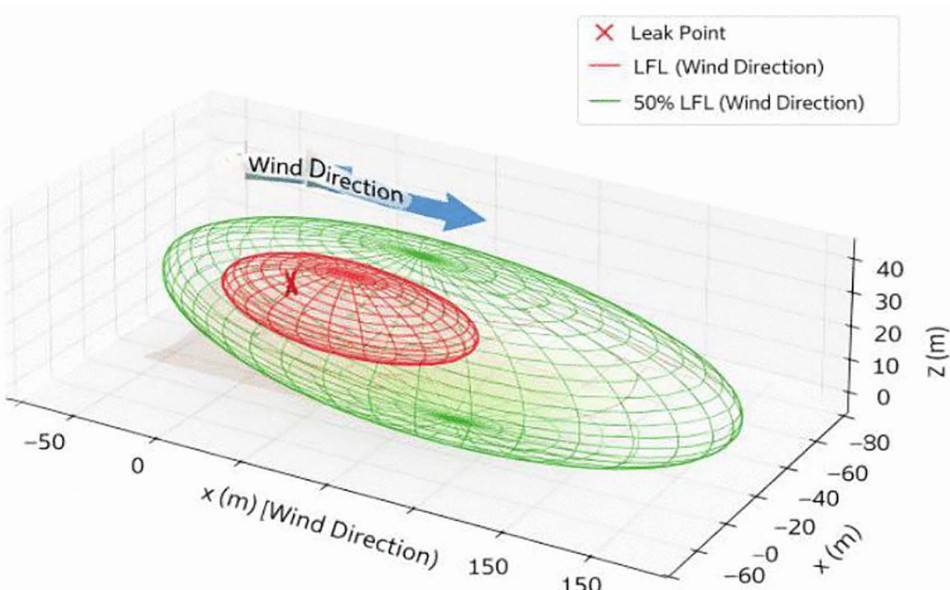

**Fig 2. Schematic 3D model of a spherical LPG storage tank with a bottom leak (leak diameter = 50 mm).**

including leak location, time of day, and seasonal period. For LFL distances, variance homogeneity was achieved after logarithmic transformation, allowing the use of one-way ANOVA followed by Tukey's HSD post hoc analysis. For 50% LFL distances, which did not meet variance homogeneity assumptions, the Kruskal–Wallis test was applied with Bonferroni-adjusted pairwise comparisons [48]. All statistical analyses were performed using Python 3.11 (SciPy, Statsmodels, and Pandas), and data visualization was carried out using Matplotlib.

The dispersion simulations were conducted under simplified assumptions to enable consistent comparison across scenarios. The surrounding terrain was assumed flat and unobstructed, and meteorological conditions were treated as steady seasonal averages rather than transient or probabilistic inputs. All scenarios were defined deterministically without repeated stochastic runs, and each simulation represents an independent observation. The analyses therefore emphasize comparative trends and scaling behavior rather than absolute worst-case dispersion distances.

Dispersion simulations were performed using PHAST v7.2, which implements the UDM. The UDM integrates jet, dense-gas, and passive dispersion regimes within a single framework and has been extensively benchmarked against large-scale dense-gas field experiments, including the Burro, Coyote, and Maplin Sands trials [49,50]. While the model relies on integral formulations and simplified turbulence treatment—potentially underestimating near-field concentration peaks compared with CFD-based models—it is widely accepted for refinery-scale consequence analysis in open and low-congestion environments, such as spherical LPG storage installations. In this study, PHAST was employed to ensure internally consistent predictions across all scenarios, with emphasis placed on comparative analysis and scaling behavior.

## Results

The Shapiro–Wilk test indicated that the dispersion distance data were not normally distributed ($W = 0.592$, $p < 0.001$). Consequently, non-parametric methods were applied for subsequent analysis. Spearman's rank correlation revealed a strong positive association between leak diameter and dispersion radius ($\rho = 0.89$, $p < 0.001$), confirming that larger leak diameters substantially increased the distance at which the gas cloud reached the LFL.

Gas composition had secondary but significant effects. A higher butane content was modestly and positively correlated with dispersion distance ($\rho = 0.36$, $p < 0.001$), while propane content showed a corresponding moderate negative correlation ($\rho = -0.36$, $p < 0.001$). In contrast, other categorical factors, including leak position (top vs. bottom), year (first vs. second half of the year), and time of day (day vs. night), had negligible associations with LFL distances ($\rho < 0.06$, $p > 0.05$). Mann–Whitney U tests supported these findings, confirming no significant differences by leak position ($U = 1716.0$, $p = 0.661$), year ($U = 1739.5$, $p = 0.753$), or time of day ($U = 1803.5$, $p = 0.987$). These results suggest that leak diameter is the dominant driver of LFL dispersion, with mixture composition having a measurable but secondary influence, while temporal and spatial factors had negligible impact ([Fig 3]).

The Shapiro–Wilk test confirmed that the 50% LFL dataset was non-normally distributed ($W = 0.604$, $p < 0.001$). Spearman's correlation revealed a strong positive association between leak diameter and 50% LFL distance ($\rho = 0.91$, $p < 0.001$), highlighting its significant role in gas dispersion. In contrast, gas composition did not show a statistically significant effect. Propane content displayed a weak-to-moderate negative correlation ($\rho = -0.32$, $p = 0.084$) while butane showed a corresponding positive correlation ($\rho = 0.32$, $p = 0.084$), but neither relationship was deemed significant. Similarly, Mann–Whitney U tests confirmed that leak position ($U = 103$, $p = 0.709$), year ($U = 1850$, $p = 0.795$), and time of day ($U = 1807$, $p = 0.973$) had no significant influence. These results suggest that dispersion distances towards the 50% LFL threshold are primarily determined by leak diameter, with minimal contributions from compositional, spatial, or temporal factors ([Fig 4]).

Across all three propane/butane mixtures, the gas dispersion distance to the LFL was consistently and substantially greater than the distance to the 50% LFL ([Fig 5]). This outcome was expected, as reaching the 50% LFL corresponds to a lower concentration threshold, resulting in a smaller flammable cloud. The longest dispersion distances, for both LFL and 50% LFL, were observed in the 15% propane/ 85% butane mixture. As the propane fraction increased and the butane

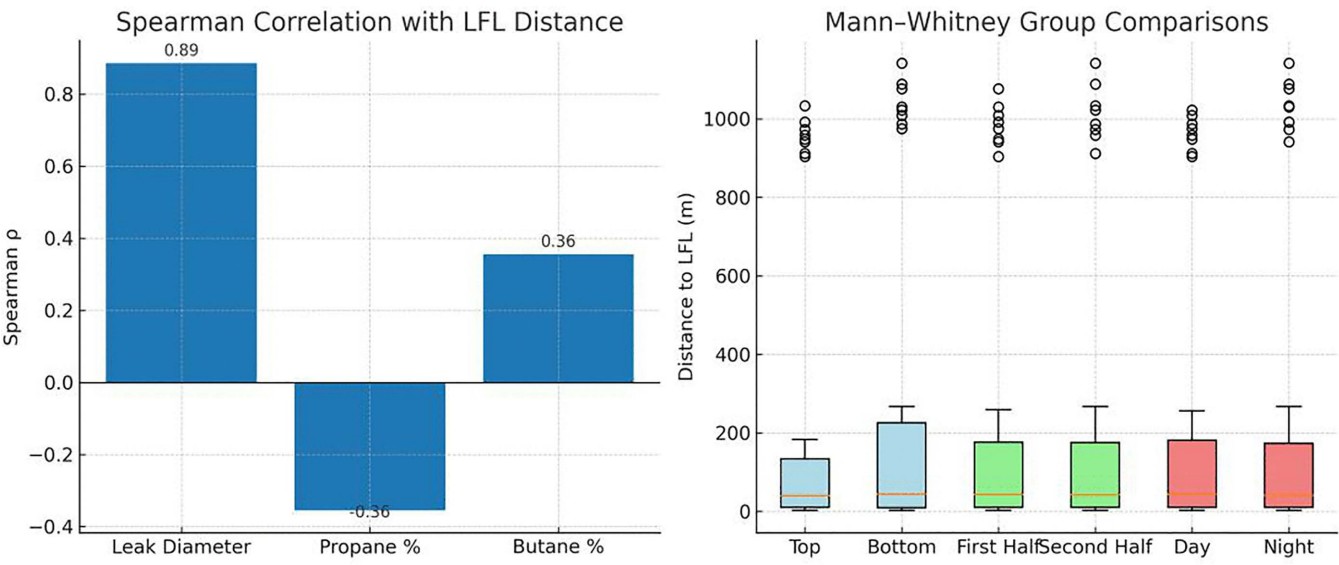

**Fig 3. Statistical analysis of dispersion distances up to the LFL.**

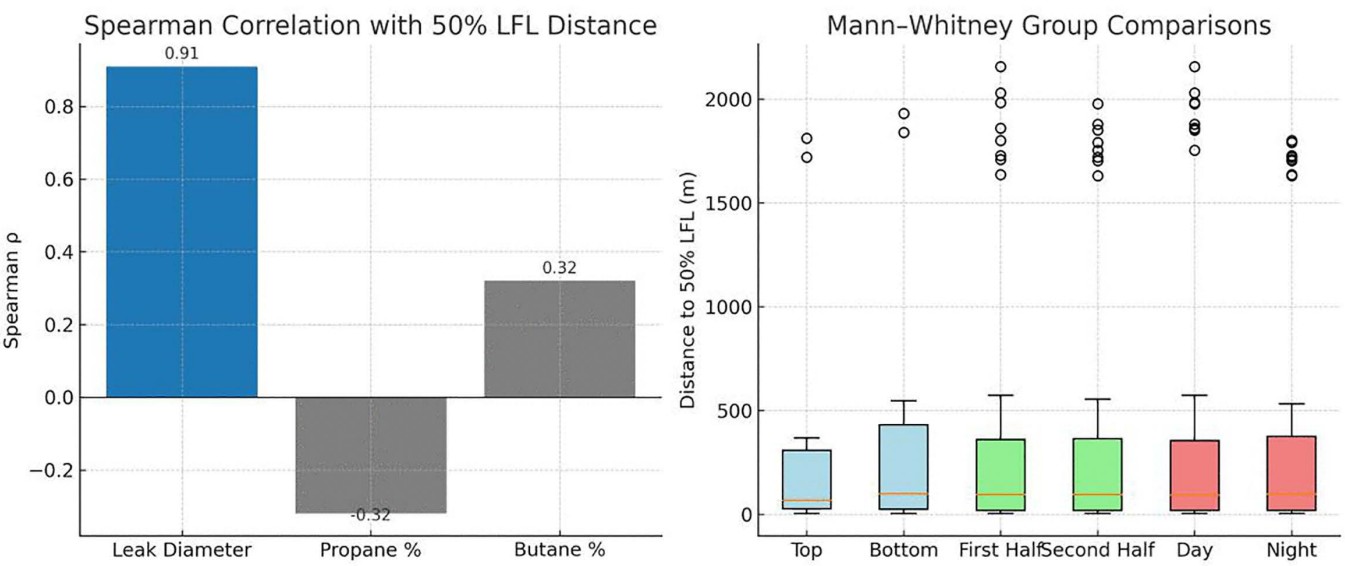

**Fig 4. Statistical analysis of dispersion distances up to the 50% LFL.**

decreased (moving towards the 50/50 mixture), dispersion distance decreased significantly. This trend suggests that butane has a stronger influence than propane in extending hazardous gas clouds. For all mixtures, the LFL distance was approximately 30–50% larger than the 50% LFL distance, with the most significant difference in the 15/85 mixture and the smallest in the 50/50 mixture. The narrow 95% confidence intervals in each group indicated low variability and high reliability of mean estimates, although slightly wider intervals in the 15/85 mixture may reflect greater meteorological variability in those scenarios. This confirms the stronger influence of butane in extending dispersion ranges.

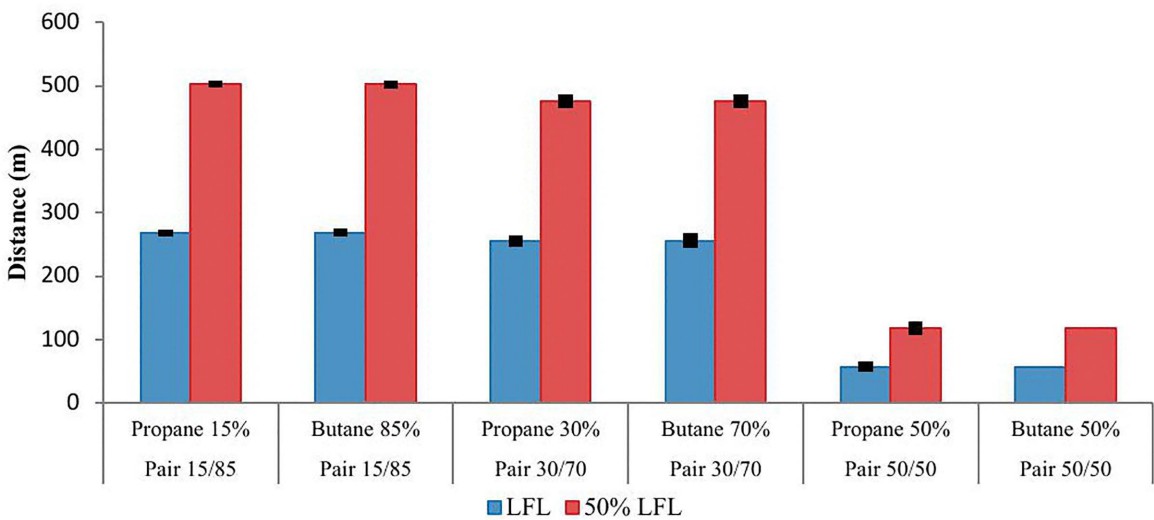

**Fig 5. Effect of LPG composition on LFL and 50% LFL dispersion distances.**

A one-way ANOVA confirmed statistically significant differences among the propane groups (F = 5.72, p = 0.004) and butane groups (F = 5.72, p = 0.004). Tukey HSD post hoc tests showed that for propane, there was no significant difference between 15% and 30% (p = 0.982), while 50% propane significantly reduced LFL distance compared to both 15% and 30% (−211.65 m and −198.78 m, respectively). For butane, both 70% and 85% mixtures significantly increased LFL distance compared to 50% (by +198.78 m and +211.65 m, respectively), with no significant difference observed between 70% and 85% (p = 0.982). These findings confirm the dominant role of butane in driving longer dispersion distances at the LFL threshold (Table 4).

For the 50% LFL threshold, the Shapiro–Wilk test once again indicated non-normality, so the Kruskal–Wallis test was applied. The results showed no statistically significant differences across propane groups (H = 3.479, p = 0.176) or butane groups (H = 3.47, p = 0.176). Pairwise Mann–Whitney tests with Bonferroni correction further confirmed the lack of

**Table 4. One-way ANOVA and Tukey HSD pairwise comparisons for propane and butane composition effects on distance to LFL distance.**

| Propane Composition – ANOVA and Tukey HSD Results | | | | | | |
|---|---|---|---|---|---|---|
| **F-statistic** | 5.722 | | | p-value | 0.004 | |
| **Group 1** | Group 2 | Mean diff | p-adj | Lower CI | Upper CI | Significant |
| **15** | 30 | −12.87 | 0.982 | −179.4 | 153.66 | — |
| **15** | 50 | −211.65 | 0.009 | −378.18 | −45.12 | ✓ |
| **30** | 50 | −198.78 | 0.015 | −365.31 | −32.25 | ✓ |
| **Butane Composition – ANOVA and Tukey HSD Results** | | | | | | |
| **F-statistic** | 5.722 | | | p-value | 0.004 | |
| **Group 1** | Group 2 | Mean diff | p-adj | Lower CI | Upper CI | Significant |
| **50** | 70 | 198.78 | 0.015 | 32.25 | 365.31 | ✓ |
| **50** | 85 | 211.65 | 0.009 | 45.12 | 378.18 | ✓ |
| **70** | 85 | 12.87 | 0.982 | −153.66 | 179.4 | — |

significant pairwise differences (p_adj > 0.05). This indicates that at sub-LFL concentrations (50% LFL), gas composition does not have a statistically significant effect on dispersion radius, with leak diameter remaining the primary factor influencing cloud size (Table 5).

The empirical relationship between leak diameter (D, mm) and gas dispersion distance to the LFL (L, m) was modeled using a power-law model and a nonlinear least-squares curve fitting method (Equation 3):

$$L = 1.22 \times D^{0.94} \tag{3}$$

The model achieved a coefficient of determination of $R^2 = 0.82$ (95% CI in log–log space). The estimated scaling exponent was $b = 0.94$ with a 95% CI of [0.86, 1.02], which includes unity. This indicates an approximately linear relationship between dispersion distance and leak diameter in log–log space. If $b < 1$, the growth rate slows slightly, indicating dissipative effects such as turbulence, buoyancy, and atmospheric entrainment. The scaling coefficient was $a = 1.22$ with a 95% CI [0.87, 1.73], representing the predicted LFL distance for a 1 mm leak under the test conditions. Fig 6 illustrates the empirical correlation compared to the normalized theoretical $D^2$ model. The empirical curve closely matched the observed data, while the $D^2$ model showed a steeper slope and tended to over-predict LFL distances at larger diameters. This deviation is consistent with dissipative mechanisms not captured in the idealized formulation.

A comparison between predicted and actual data was conducted to assess the accuracy of the model, as shown in Fig 7. The 1:1 reference line (y = x) signifies perfect agreement between predictions and actual results. The regression, with a slope of approximately 1 (m ≈ 1) and an intercept close to 0 (c ≈ 0), indicates no significant bias in the model. The coefficient of determination ($R^2 = 0.82$) indicates that the model explains roughly 82% of the variability in LFL distances. Overall, the empirical power-law model demonstrated strong predictive performance, showing almost linear scaling with leak diameter, and closely aligning with observed data within the tested range.

The empirical relationship between leak diameter (D, mm) and gas dispersion distance to the 50% LFL (L, m) was modeled using a nonlinear least-squares power-law model (Equation 4):

$$L = 2.10 \times D^{0.96} \tag{4}$$

The model achieved a coefficient of determination of $R^2 = 0.83$ (95% CI in log–log space). The estimated scaling exponent was $b = 0.96$ with a 95% CI of [0.88, 1.04], including unity. This indicates an approximately linear dependence of 50% LFL distance on leak diameter in log–log space. If $b < 1$, the growth rate is modestly decelerating, consistent with dissipative processes such as turbulence, buoyancy, and atmospheric entrainment. The scaling coefficient was $a = 2.10$ with a 95%

**Table 5. Results of pairwise comparisons for the effect of propane and butane composition on distance to 50% LFL.**

| Propane Composition | | | | | | |
|---|---|---|---|---|---|---|
| H-statistic | 5.722 | | | p-value | 0.176 | |
| Group 1 | Group 2 | Mean diff | p-adj | Lower CI | Upper CI | Significant |
| 15 | 30 | 26.7 | 1.0 | NA | NA | — |
| 15 | 50 | 384.66 | 0.364 | NA | NA | — |
| 30 | 50 | 357.96 | 0.364 | NA | NA | — |
| Butane Composition – Kruskal–Wallis Results | | | | | | |
| H-statistic | 3.479 | | | p-value | 0.176 | |
| Group 1 | Group 2 | Mean diff | p-adj | Lower CI | Upper CI | Significant |
| 50 | 70.0 | −357.96 | 0.364 | NA | NA | — |
| 50 | 85.0 | −384.66 | 0.364 | NA | NA | — |
| 70 | 85.0 | −26.7 | 1.0 | NA | NA | — |

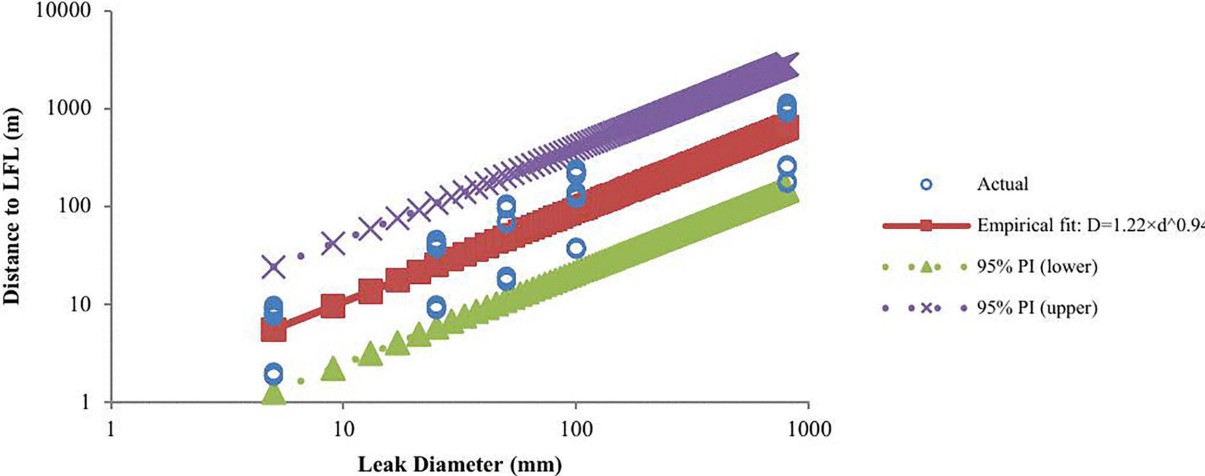

**Fig 6. LFL – Comparison of empirical and theoretical models with actual data (log–log axes).**

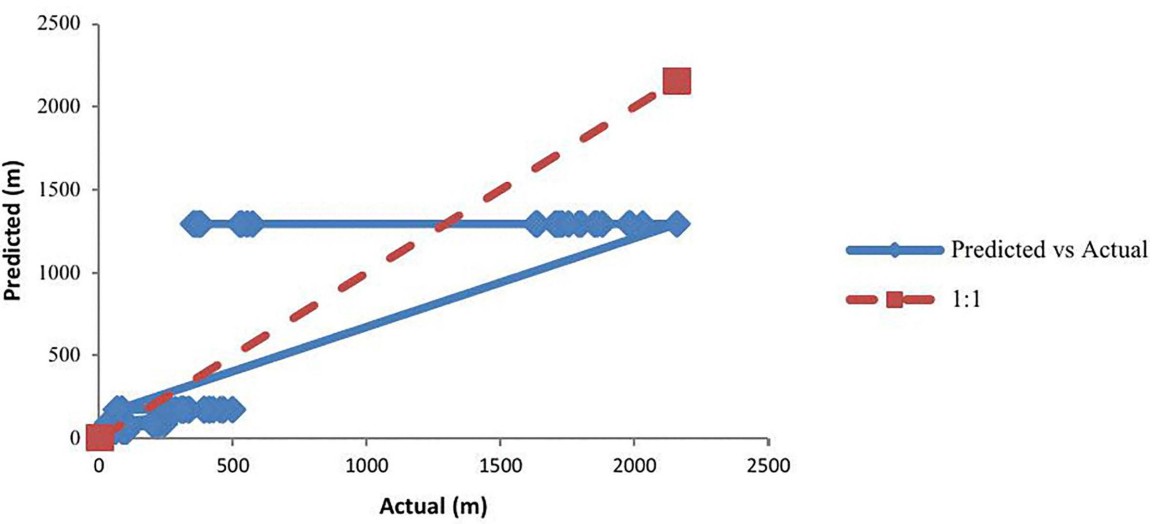

**Fig 7. Predicted vs. actual LFL distances with 1:1 reference line and fitted regression.**

CI of [1.50, 2.96], representing the predicted 50% LFL distance for a 1 mm leak under the test conditions. Fig 8 compares the empirical correlation with the normalized theoretical $D^2$ model. The empirical curve closely tracked the observed data, while the $D^2$ model exhibited a steeper slope and tended to over-predict distances at larger diameters, reflecting the influence of physical dissipative effects not accounted for in the theoretical formulation.

The model's performance was evaluated using a predicted versus actual plot (Fig 9). The 1:1 reference line ($y = x$) represents perfect agreement between predictions and observations. The fitted regression line had a slope of $m \approx 1$ and an intercept of $c \approx 0$, indicating no statistically significant bias. With a coefficient of determination ($R^2 = 0.83$), the model explained approximately 83% of the variance in measured 50% LFL distances. Overall, the empirical model demonstrated strong predictive capabilities, showing nearly linear scaling with leak diameter, and consistently agreeing with observed data within the tested range.

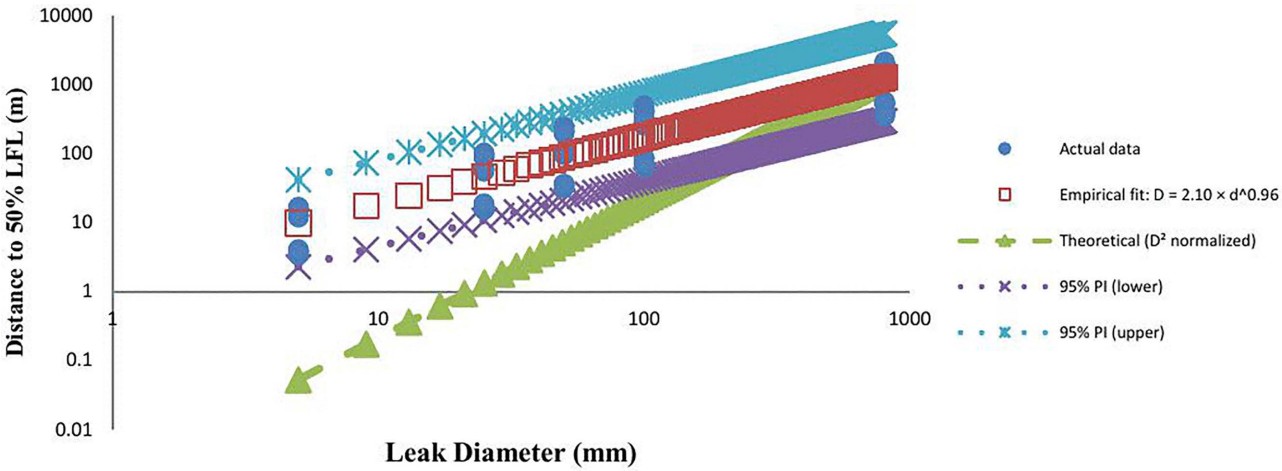

**Fig 8. 50% LFL – Comparison of empirical and theoretical models with actual data (log–log axes).**

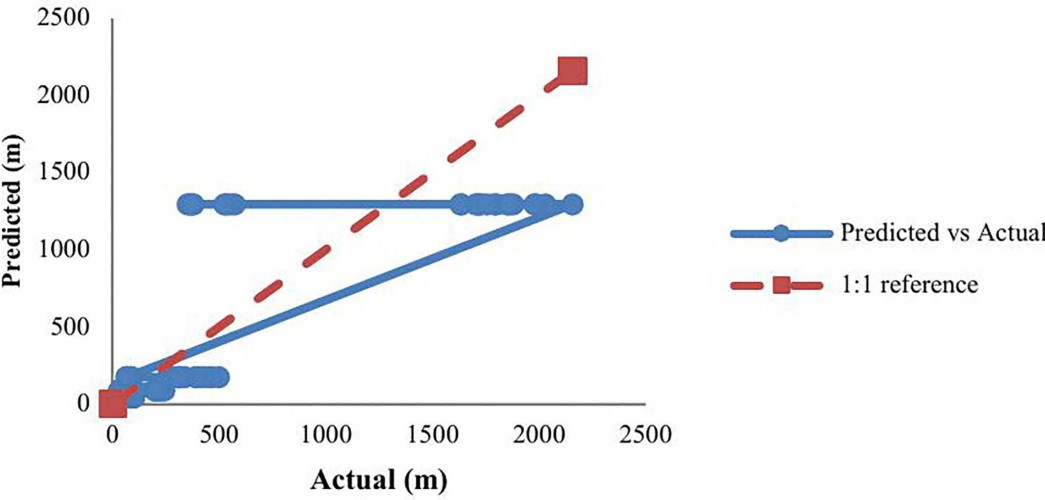

**Fig 9. Predicted vs Actual 50% LFL distances with 1:1 reference line and fitted regression.**

The empirical power-law correlations for both LFL and 50% LFL dispersion distances yield scaling exponents close to unity (LFL: b = 0.94, 95% CI [0.86–1.02]; 50% LFL: b = 0.96, 95% CI [0.88–1.04]). This near-linear log–log relationship indicates that the influence of leak diameter on dispersion distance is consistent across concentration thresholds. However, the baseline coefficient "a" is significantly higher for the 50% LFL endpoint (a = 2.10, 95% CI [1.50–2.96]) compared to the LFL (a = 1.22, 95% CI [0.87–1.73]), reflecting the greater distance required to reach the lower concentration limit. Both models exhibit strong performance $(R^2 \approx 0.82–0.83)$ and 95% prediction intervals confirm their reliability for estimating dispersion distances under similar conditions. Fig 10 shows a log–log comparison of empirical power-law fits for LFL and 50% LFL endpoints against a theoretical $D^2$ model. The 50% LFL curve is vertically shifted upward, reflecting the increased distance necessary to reach the lower flammability threshold, while both curves maintain near-linear scaling with leak diameter.

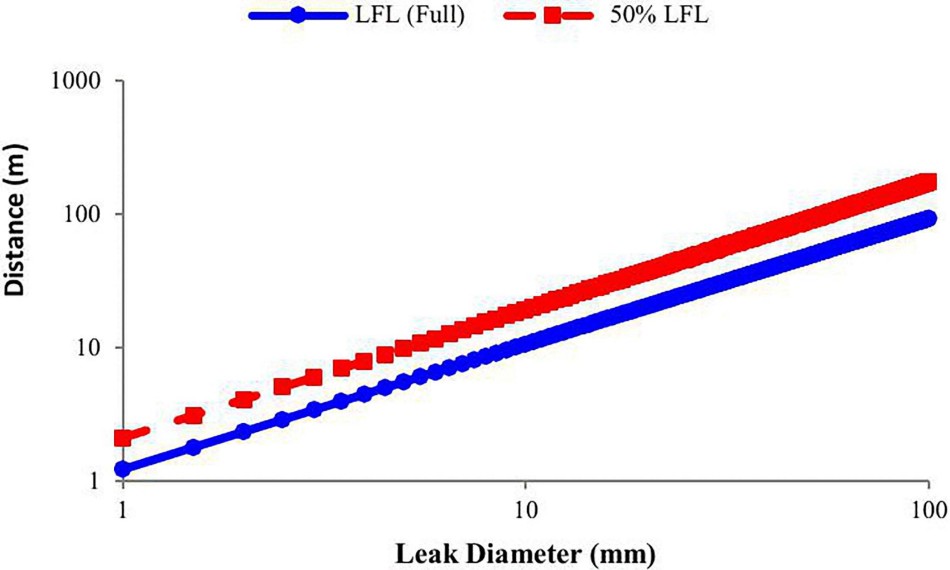

**Fig 10. LFL and 50% LFL dispersion distance in compassion together.**

## Discussion

### Interpretation of results

This study demonstrates that the diameter of the leak is the main factor influencing the dispersion of LPG vapor clouds in refinery-scale conditions. Strong correlations were found between the leak diameter and dispersion distance for both concentration thresholds ($\rho = 0.89$ for LFL and $\rho = 0.91$ for 50% LFL, $p < 0.001$). The Power-law regression showed almost linear scaling in log–log space ($b = 0.94$ for LFL and $b = 0.96$ for 50% LFL), indicating that dispersion distance increases almost proportionally with leak size. Similar relationships between the size of the source and the hazard distance have been observed in experimental and numerical studies of dense-gas releases [22,23].

The consistently greater distances associated with the 50% LFL endpoint confirm its conservative nature for safety applications, in line with recommendations from NFPA (2019) [8] and API (2009) [7]. While the composition of LPG influenced LFL distances, especially for butane-rich mixtures, its impact decreased at the 50% LFL threshold, where geometric scaling was more prominent. Similar trends have been observed in CFD-based analyses of LPG and LNG dispersion, where compositional effects weaken at lower concentration thresholds due to enhanced dilution [5,51]. In the specific conditions of the site examined in this study, factors such as leak location, season, and time of day did not result in statistically significant differences, consistent with previous refinery-scale consequence studies that used averaged meteorological data [4,22,52].

### Physical interpretation and scaling behavior

The observed scaling behavior aligns with fundamental fluid mechanics principles. According to classical orifice flow theory [53–56], the volumetric release rate from a pressurized vessel scales with the square of the orifice diameter and the square root of the pressure differential ($Q \propto D^2 \sqrt{\Delta P}$). With a relatively constant operating pressure, larger leaks are expected to produce larger vapor clouds that disperse over greater distances. However, the slightly sub-linear scaling exponents found in this study (0.94–0.96) indicate the presence of dissipative mechanisms that limit cloud expansion. Turbulence-enhanced mixing, buoyancy-induced plume rise, and atmospheric entrainment all play a role in dilution and

decrease downwind distances, especially for significant releases. These mechanisms have been extensively examined in dense-gas field experiments such as the Burro and Maplin Sands trials [15,57,58]; and in CFD simulations of industrial gas releases [10,22,23,59].

## Comparison with previous studies

The present findings are consistent with previous experimental and numerical research on dense-gas dispersion. Large-scale field trials, such as the Burro and Maplin Sands experiments, have shown a clear correlation between dispersion distance and source size, as well as the influence of atmospheric turbulence and entrainment [15,57,58]. CFD studies by Pontiggia et al. (2011) [22] and Lozano (2024) [10] have indicated that simple D² scaling laws may overestimate dispersion distances for larger leaks, indicating that empirical or regression-based correlations provide a more accurate representation of actual behavior.

Recent studies have expanded on these findings in various contexts. Le et al. (2023) [5] highlighted the importance of leak size and wind speed for small LPG releases, while Lozano (2024) [10] illustrated how obstacles and road geometry can influence dispersion patterns in tanker accidents. Research in confined or urban environments has emphasized the importance of ventilation, congestion, and gas composition [38,51,52]. In comparison to this body of literature, the present study specifically investigates refinery-scale spherical LPG storage tanks and offers a systematic comparison between LFL and 50% LFL endpoints under consistent operational and meteorological conditions, a topic that has not been explored in existing literature.

## Practical implications, limitations, and future work

From a process safety perspective, the results indicate that hazard zoning and land-use planning should prioritize leak diameter when defining consequence distances. This approach aligns with the guidance provided by CCPS (2011) [4] and API (2009) [7]. Setting the design boundary at 50% LFL offers an additional safety margin against turbulent concentration fluctuations and localized ignition, as recommended by NFPA (2019) [8]. It is also important to consider seasonal variations in LPG composition, especially increased butane content, when establishing safety distances, as supported by recent numerical studies [5,37].

Several limitations should be acknowledged. The analysis relies on PHAST's Unified Dispersion Model, which employs integral formulations and simplified turbulence treatment and may underestimate near-field concentration peaks compared with CFD models [15,22,60]. Meteorological inputs were represented using averaged conditions rather than time-resolved or probabilistic data, and terrain complexity and congestion effects were not considered. Future research should integrate CFD-based modeling, transient meteorology, and probabilistic frameworks to better capture near-field plume behavior and uncertainty, as suggested by recent hybrid CFD–data-driven studies [52].

## Conclusion

This study confirms that the diameter of the leak is the primary factor governing LPG vapor cloud dispersion from pressurized spherical storage tanks. It shows nearly linear power-law scaling at both the LFL and 50% LFL thresholds. Butane-rich mixtures increase dispersion distances at the LFL, while compositional effects diminish at the 50% LFL, where geometric scaling becomes dominant. Under the site-specific conditions examined, meteorological and temporal variations did not produce statistically significant differences, highlighting the dominant role of leak size and fuel composition in determining hazard distances. The empirical correlations developed provide practical support for hazard zoning, consequence analysis, and emergency planning in refinery environments. Adopting the 50% LFL as a conservative design boundary enhances safety margins by accounting for turbulent concentration fluctuations and localized ignition risks. Considering seasonal LPG composition can further refine safety distances. The analysis is subject to limitations associated with the use of integral dispersion modeling and averaged meteorological inputs. Future work should therefore integrate

high-resolution CFD simulations and probabilistic meteorological frameworks to better capture near-field plume behavior and uncertainty. Overall, the findings support the use of dual flammability thresholds as a consistent and conservative basis for safety-distance determination in LPG storage facilities.

## Author contributions

**Conceptualization:** Keyvan Sarebanzadeh, Mahboubeh Es'haghi.

**Data curation:** Keyvan Sarebanzadeh, Mahboubeh Es'haghi.

**Formal analysis:** Keyvan Sarebanzadeh, Mahboubeh Es'haghi.

**Methodology:** Keyvan Sarebanzadeh, Mahboubeh Es'haghi.

**Software:** Keyvan Sarebanzadeh, Mahboubeh Es'haghi.

**Supervision:** Mahboubeh Es'haghi.

**Visualization:** Mahboubeh Es'haghi.

**Writing – original draft:** Keyvan Sarebanzadeh, Mahboubeh Es'haghi.

**Writing – review & editing:** Mahboubeh Es'haghi.

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
