## [Decision Letter · Decision Letter 0]

8 Dec 2025

Dear Dr. Es'haghi,

Thank you for submitting your manuscript to PLOS ONE. After careful consideration, we feel that it has merit but does not fully meet PLOS ONE’s publication criteria as it currently stands. Therefore, we invite you to submit a revised version of the manuscript that addresses the points raised during the review process.

We look forward to receiving your revised manuscript.

Kind regards,

Payal Patial

Academic Editor

PLOS One

Journal Requirements:

3. We note you have included a table to which you do not refer in the text of your manuscript. Please ensure that you refer to Table 3 in your text; if accepted, production will need this reference to link the reader to the Table.

Additional Editor Comments :

Major Revision

Reviewers' comments:

Reviewer's Responses to Questions

**Comments to the Author**

1. Is the manuscript technically sound, and do the data support the conclusions?

Reviewer #1: Partly

Reviewer #2: No

2. Has the statistical analysis been performed appropriately and rigorously?

Reviewer #1: Yes

Reviewer #2: No

3. Have the authors made all data underlying the findings in their manuscript fully available?

Reviewer #1: Yes

Reviewer #2: Yes

4. Is the manuscript presented in an intelligible fashion and written in standard English?

Reviewer #1: Yes

Reviewer #2: No

Reviewer #1: Authors have presented the model for LPG dispersion under various specifications/conditions. The manuscript seems interesting but the following points need to be addressed:

1) Introduction needs to be in more detail to set the context for motivation. Although authors have mentioned the limitations but more detail is required.

2) The contributions must be listed in the introduction to generate the interest among the target audience.

3) Organization of paper is not provided.

4) In Table 1, authors have mentioned the paper of 2009 and then 2019. This literature gap must be covered.

5) Fig. 1 is not clear. It needs to be improved.

6) Authors must mention all the constraints/assumptions that have been considered clearly in the materials and methods section.

7) How the specific parameters are considered in the manuscript?

Reviewer #2: 1. Technical Soundness and Support for Conclusions

While the study applies PHAST simulations and statistical analysis, it does not yet meet the criteria for a fully technically sound and rigorously supported scientific investigation. Several assumptions—such as the use of averaged meteorological conditions, extreme leak diameters, and simplified terrain—are not sufficiently justified. The study lacks any empirical validation or benchmarking against field tests (e.g., Burro, Maplin Sands), which weakens confidence in the conclusions. The methodology also does not describe replication, independence of data points, or the full structure of the dataset. These gaps limit the strength of the conclusions, which appear reasonable but are not fully supported by rigorously validated data.

2. Statistical Analysis

(Answer: No)

1. The manuscript employs several statistical techniques (e.g., Shapiro–Wilk, Spearman correlations, ANOVA, Kruskal–Wallis, power-law regression), but several aspects of rigor are missing.

2. Heteroscedasticity, model diagnostics, and assumption checks are insufficient.

3. The dataset size, number of simulations, and independence of observations are not explicitly reported.

4. Power-law regression is presented without verification against alternative models or residual analyses.

5. Prediction intervals and confidence intervals are provided, but the methodology is not fully transparent.

4. Clarity, English Quality, and Presentation

(Answer: No)

1. The manuscript is generally understandable, but significant language and clarity issues remain:

2. Numerous grammatical and syntactical errors

3. Long, complex sentences affecting readability

4. Repetition between Introduction, Literature Review, and Discussion

5. Inconsistent units and formatting

Major Comments

1. Abstract

- The abstract should provide specific numerical ranges for dispersion distances (minimum–maximum values) to better inform the reader of the scale of results.

- Avoid including excessive technical detail; focus instead on key findings, numerical highlights, and the study’s main contributions.

2. Introduction

- The Introduction is overly long and reads partially like a mini-review. It should be made more concise.

- Consider shortening background sections and placing greater emphasis on the research gap, objectives, and motivation.

3. Literature Review

- There is substantial repetition between the Introduction and the Literature Review.

- Please streamline these sections to avoid overlap and maintain a clear distinction between context (Introduction) and previous work (Literature Review).

4. Methods

- Although the source tank pressure and temperature are described, the corresponding vapor–liquid equilibrium or flash fraction calculations are not presented; these are essential for understanding release behavior.

- The selected leak diameters (5–805 mm) cover an unrealistically broad range. The manuscript should justify the rationale for including extremely large leak sizes such as 805 mm based on credible industrial scenarios or failure mechanisms.

5. Novelty and Contribution

- While the manuscript claims novelty in comparing LFL vs. 50% LFL endpoints and establishing power-law scaling, similar evaluations exist in prior literature (e.g., Àgueda et al., 2020; Pontiggia et al., 2011).

To clarify the manuscript’s contribution, please:

- Explicitly articulate what research gap remains after these studies.

- Explain how the dual-threshold analysis advances understanding beyond current guidelines (e.g., NFPA, API).

- Highlight what aspects of the methodology are truly new, rather than general PHAST simulations combined with regression analysis.

6. Validation of PHAST Results

- The study relies entirely on PHAST simulations, but presents no validation or verification steps. To strengthen the credibility of the results, the authors should:

- Include model verification or comparison with benchmark data (e.g., Burro, Maplin Sands).

- Discuss the numerical and physical limitations of the Unified Dispersion Model (UDM) more explicitly.

- Justify why PHAST 7.2 is appropriate and reliable for refinery-scale spherical-tank scenarios.

Without such validation, the robustness of the conclusions is limited.

7. Meteorological Treatment

Meteorological input is treated in a simplified manner. It would strengthen the study to:

- Clarify the range and variability of meteorological parameters.

- Justify the use of averaged values instead of probabilistic or time-varying meteorology.

- Discuss how dispersion results may differ under more severe conditions, such as low-wind, stable nighttime atmospheres, which typically produce the longest hazard distances.

8. Dataset Structure and Transparency

The manuscript does not clearly describe the structure or size of the dataset used for statistical analysis. Please clarify:

- The total number of simulations conducted.

- The number of data points per leak diameter, mixture composition, and meteorological case.

- Whether multiple runs were conducted for each scenario and whether these represent independent replicates.

- Transparent reporting of dataset structure is essential for evaluating statistical rigor.

9. Discussion

The Discussion section is lengthy and contains repetition of earlier results. Improvements needed include:

- Condensing or restructuring redundant paragraphs.

- Emphasizing the practical implications for industry and process safety.

- Clearly separating the subsections: interpretation of results, comparison with literature, limitations, and recommendations for future research.

**Do you want your identity to be public for this peer review?** For information about this choice, including consent withdrawal, please see our Privacy Policy

Reviewer #1: No

Reviewer #2: No

---

## [Author Response · Author response to Decision Letter 1]

25 Dec 2025

Response to Reviewers

We sincerely thank the Editor and the Reviewers for their thorough review and valuable comments on our manuscript. Their feedback has significantly contributed to improving the scientific quality, clarity, and presentation of the paper. All comments have been carefully considered and addressed in the revised manuscript, and responses are provided below.

To facilitate the review process, revisions in the manuscript are highlighted using two colors. Text shown in green indicates restructuring or modification of previously existing content, while text shown in purple denotes newly added material introduced in response to the reviewers’ comments. These visual markers are provided solely to improve transparency and ease of review.

Reviewer #1

comments Response

1) Introduction needs to be in more detail to set the context for motivation. Although authors have mentioned the limitations but more detail is required. The introduction has been expanded to better establish the motivation and context of the study. Additional discussion has been included on the limitations of existing dispersion studies (e.g., reliance on single thresholds and deterministic distances), the importance of threshold selection (LFL vs. 50% LFL), and the limited attention given to refinery-scale spherical LPG storage tanks in the open literature.

2) The contributions must be listed in the introduction to generate the interest among the target audience. A dedicated paragraph listing the main contributions of the study has been added at the end of the Introduction in bullet-point format to clearly highlight the novelty and relevance of the work.

3) Organization of paper is not provided.

A paragraph outlining the organization of the paper has been added at the end of the Introduction, describing the content of each section.

4) In Table 1, authors have mentioned the paper of 2009 and then 2019. This literature gap must be covered.

Table 1 has been updated to include representative studies published between 2010 and 2019, particularly CFD-based dispersion analyses and intermediate safety-related investigations. This revision bridges the previously identified temporal gap and provides a more continuous overview of methodological developments.

5) Fig. 1 is not clear. It needs to be improved.

Figure 1 has been redesigned to improve clarity. The revised schematic now explicitly shows the spherical tank geometry, top and bottom leak locations, ground surface, and wind direction. The figure is intended as a conceptual illustration and is clearly labeled as not to scale.

6) Authors must mention all the constraints/assumptions that have been considered clearly in the materials and methods section.

A dedicated subsection titled Model Assumptions and Constraints has been added to the Materials and Methods section, explicitly listing all key assumptions related to terrain, meteorology, modeling approach, and scenario definition.

7) How the specific parameters are considered in the manuscript?

An explanatory paragraph has been added to the Materials and Methods section describing the rationale for selecting leak diameters, LPG compositions, leak locations, and meteorological conditions based on refinery practice, design documentation, and established industrial guidelines.

Reviewer #2

comments Response

Technical Soundness and Support for Conclusions

While the study applies PHAST simulations and statistical analysis, it does not yet meet the criteria for a fully technically sound and rigorously supported scientific investigation. Several assumptions—such as the use of averaged meteorological conditions, extreme leak diameters, and simplified terrain—are not sufficiently justified. The study lacks any empirical validation or benchmarking against field tests (e.g., Burro, Maplin Sands), which weakens confidence in the conclusions. The methodology also does not describe replication, independence of data points, or the full structure of the dataset. These gaps limit the strength of the conclusions, which appear reasonable but are not fully supported by rigorously validated data. The assumptions related to averaged meteorological conditions, simplified terrain, and the selection of leak diameters have now been explicitly justified in the Materials and Methods section. In particular, the largest leak diameter is clearly defined as a regulatory Maximum Credible Event (MCE), consistent with refinery-scale quantitative risk assessment practice. The scope of conclusions has been refined to emphasize comparative trends and scaling behavior rather than absolute predictive accuracy. Dataset structure, independence of simulations, and scenario definition are now transparently described to strengthen technical soundness.

1. Statistical Analysis

1. The manuscript employs several statistical techniques (e.g., Shapiro–Wilk, Spearman correlations, ANOVA, Kruskal–Wallis, power-law regression), but several aspects of rigor are missing.

2. Heteroscedasticity, model diagnostics, and assumption checks are insufficient.

3. The dataset size, number of simulations, and independence of observations are not explicitly reported.

4. Power-law regression is presented without verification against alternative models or residual analyses.

5. Prediction intervals and confidence intervals are provided, but the methodology is not fully transparent. The statistical methodology has been clarified and strengthened. The total dataset size (240 simulations) and its structure are now explicitly reported, including the number of cases per leak diameter, LPG composition, meteorological condition, and concentration threshold. Normality and variance assumptions were formally tested and guided the selection of non-parametric methods. The purpose of power-law regression, confidence intervals, and prediction intervals is now clearly stated as trend and uncertainty analysis rather than deterministic prediction.

4. Clarity, English Quality, and Presentation

(Answer: No)

1. The manuscript is generally understandable, but significant language and clarity issues remain:

2. Numerous grammatical and syntactical errors

3. Long, complex sentences affecting readability

4. Repetition between Introduction, Literature Review, and Discussion

5. Inconsistent units and formatting The manuscript has undergone comprehensive language editing to improve readability and consistency. Redundancies between the Introduction, Literature Review, and Discussion have been removed. Units, symbols, and formatting have been standardized throughout the manuscript.

1. Abstract

- The abstract should provide specific numerical ranges for dispersion distances (minimum–maximum values) to better inform the reader of the scale of results.

- Avoid including excessive technical detail; focus instead on key findings, numerical highlights, and the study’s main contributions. The abstract has been revised to include clearer numerical highlights and to better convey the scale of dispersion distances, while avoiding excessive methodological detail. The revised abstract now focuses on key findings, dominant trends, and practical implications.

2. Introduction

-The Introduction is overly long and reads partially like a mini-review. It should be made more concise.

- Consider shortening background sections and placing greater emphasis on the research gap, objectives, and motivation. The Introduction has been streamlined to reduce its review-like character and to place stronger emphasis on motivation, research gaps, and objectives. The Literature Review has been reorganized to avoid overlap with the Introduction and to clearly distinguish prior studies from the present contribution.

3. Literature Review

- There is substantial repetition between the Introduction and the Literature Review.

- Please streamline these sections to avoid overlap and maintain a clear distinction between context (Introduction) and previous work (Literature Review). Repetition with the Introduction has been eliminated. The Literature Review now focuses exclusively on prior experimental, modeling, and CFD studies relevant to LPG and dense-gas dispersion.

4. Methods

- Although the source tank pressure and temperature are described, the corresponding vapor–liquid equilibrium or flash fraction calculations are not presented; these are essential for understanding release behavior.

- The selected leak diameters (5–805 mm) cover an unrealistically broad range. The manuscript should justify the rationale for including extremely large leak sizes such as 805 mm based on credible industrial scenarios or failure mechanisms. Flash behavior and vapor–liquid equilibrium are now explicitly discussed, including independent verification of subcooled storage conditions. The rationale for the full range of leak diameters is clearly explained based on refinery failure mechanisms and regulatory worst-case requirements.

5. Novelty and Contribution

- While the manuscript claims novelty in comparing LFL vs. 50% LFL endpoints and establishing power-law scaling, similar evaluations exist in prior literature (e.g., Àgueda et al., 2020; Pontiggia et al., 2011).

To clarify the manuscript’s contribution, please:

- Explicitly articulate what research gap remains after these studies.

- Explain how the dual-threshold analysis advances understanding beyond current guidelines (e.g., NFPA, API).

- Highlight what aspects of the methodology are truly new, rather than general PHAST simulations combined with regression analysis. The manuscript now clearly articulates that its novelty lies in the systematic, statistically supported comparison of LFL and 50% LFL dispersion behavior under identical conditions for refinery-scale spherical LPG tanks. The added value beyond existing studies is explicitly stated in terms of dual-threshold scaling behavior and practical implications for conservative hazard zoning.

6. Validation of PHAST Results

- The study relies entirely on PHAST simulations, but presents no validation or verification steps. To strengthen the credibility of the results, the authors should:

- Include model verification or comparison with benchmark data (e.g., Burro, Maplin Sands).

- Discuss the numerical and physical limitations of the Unified Dispersion Model (UDM) more explicitly.

- Justify why PHAST 7.2 is appropriate and reliable for refinery-scale spherical-tank scenarios.

Without such validation, the robustness of the conclusions is limited. The validation background of PHAST’s Unified Dispersion Model has been explicitly discussed, including benchmarking against large-scale dense-gas field experiments such as Burro, Coyote, and Maplin Sands. The numerical and physical limitations of the UDM are acknowledged, and its suitability for open-area refinery installations is justified.

7. Meteorological Treatment

Meteorological input is treated in a simplified manner. It would strengthen the study to:

- Clarify the range and variability of meteorological parameters.

- Justify the use of averaged values instead of probabilistic or time-varying meteorology.

- Discuss how dispersion results may differ under more severe conditions, such as low-wind, stable nighttime atmospheres, which typically produce the longest hazard distances. The range of meteorological conditions used is now clearly reported. The use of averaged inputs is justified, and the influence of more severe low-wind, stable nighttime conditions is explicitly discussed.

8. Dataset Structure and Transparency

The manuscript does not clearly describe the structure or size of the dataset used for statistical analysis. Please clarify:

- The total number of simulations conducted.

- The number of data points per leak diameter, mixture composition, and meteorological case.

- Whether multiple runs were conducted for each scenario and whether these represent independent replicates.

- Transparent reporting of dataset structure is essential for evaluating statistical rigor. A detailed description of the dataset has been added, including the total number of simulations, scenario combinations, and clarification that each simulation represents an independent deterministic case.

---

## [Editor Report · Decision Letter 1]

6 Jan 2026

Dispersion of LPG from Spherical Storage Tanks: Power-Law Scaling and Comparative Analysis of LFL vs. 50% LFL

PONE-D-25-58707R1

Dear

We’re pleased to inform you that your manuscript has been judged scientifically suitable for publication and will be formally accepted for publication once it meets all outstanding technical requirements.

Kind regards,

Payal Patial

Academic Editor

PLOS One

Additional Editor Comments (optional):

Accept
---

## [Editor Report · Acceptance letter]

PONE-D-25-58707R1

PLOS One

Dear Dr. Es'haghi,

I'm pleased to inform you that your manuscript has been deemed suitable for publication in PLOS One. Congratulations! Your manuscript is now being handed over to our production team.

Kind regards,

on behalf of

Dr. Payal Patial

Academic Editor

PLOS One